# Cyclic Contrastive Knowledge Transfer for Open-Vocabulary Object Detection

**Chuhan Zhang[1,2], Chaoyang Zhu[1], Pingcheng Dong[1], Long Chen[1], Dong Zhang[1,2*]**
[1]The Hong Kong University of Science and Technology
[2]AI Chip Center for Emerging Smart Systems (ACCESS)
{chuhanzhang,longchen,dongz}@ust.hk;sean.zhuh@gmail.com
pingcheng.dong@connect.ust.hk

## Abstract

In pursuit of detecting unstinted objects that extend beyond predefined categories, prior arts of open-vocabulary object detection (OVD) typically resort to pretrained vision-language models (VLMs) for base-to-novel category generalization. However, to mitigate the misalignment between upstream image-text pretraining and downstream region-level perception, additional supervisions are indispensable, *e.g.*, image-text pairs or pseudo annotations generated via self-training strategies. In this work, we propose CCKT-Det trained *without* any extra supervision. The proposed framework constructs a cyclic and dynamic knowledge transfer from language queries and visual region features extracted from VLMs, which forces the detector to closely align with the visual-semantic space of VLMs. Specifically, 1) we prefilter and inject semantic priors to guide the learning of queries, and 2) introduce a regional contrastive loss to improve the awareness of queries on novel objects. CCKT-Det can consistently improve performance as the scale of VLMs increases, all while requiring the detector at a moderate level of computation overhead. Comprehensive experimental results demonstrate that our method achieves performance gain of $+2.9\%$ and $+10.2\%$ $AP_{50}$ over previous state-of-the-arts on the challenging COCO benchmark, both *without* and *with* a stronger teacher model. The code is provided at https://github.com/ZCHUHan/CCKT-Det.

## 1 Introduction

Object detection, a fundamental perception task in computer vision, entails both localization and classification of objects on given images. The last decade has witnessed great success in object detection including advanced architectures (Ren, 2015; Carion et al., 2020), tailored loss objectives (Lin et al., 2017; Rezatofighi et al., 2019), and large-scale datasets (Gupta et al., 2019; Shao et al., 2019). However, canonical close-set detectors are constrained by small-scale and predefined categories, rendering them less capable of detecting new concepts (Bansal et al., 2018; Gu et al., 2021). To deploy detectors in real-world scenarios with countless concepts, many of which have no annotations in the training set, open-vocabulary object detection is formulated to facilitate the recognition of novel objects (Zareian et al., 2021).

Drawing on the impressive image-level zero-shot learning capabilities of vision-language models like CLIP (Radford et al., 2021), recent initiatives replace the learnable classifier weights in traditional detectors with frozen text embeddings generated from VLMs text encoder that takes as input template prompts filled with category names. To address the challenges posed by the misalignment between image-text pre-training and region-level perception, as shown in Figure 1, existing methods heavily rely on additional data such as: 1) caption datasets, *e.g.*, CC3M (Sharma et al., 2018) and COCO Caption (Chen et al., 2015), or 2) meticulously designed self-training strategies to generate pseudo annotations. These extra supervisions are typically coarse and noisy, leading to inaccurate region-word alignment. This raises the question of whether a moderate level of weak supervision is sufficient while effectively maximizing the knowledge transfer from VLMs and multi-modal large language models (MLLMs) (Wang et al., 2023).

---

*Corresponding author (Dong Zhang).

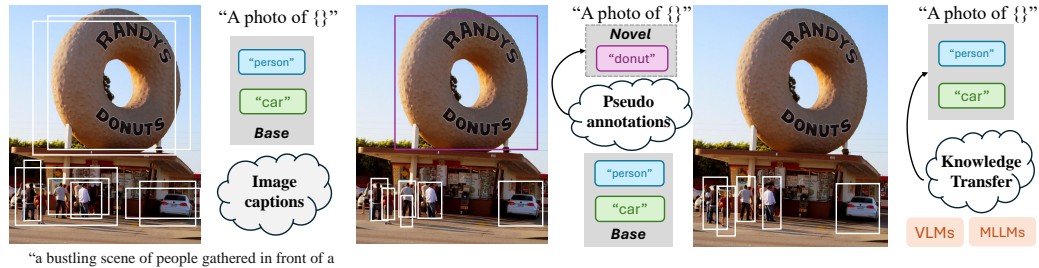

Figure 1: The evolution of extracting novel concepts in OVD models. Compared to existing methods by using extra captions in (a) or pseudo annotations & self-training strategies in (b), we propose leveraging semantic priors to reveal novel concepts and employing contrastive knowledge distillation paradigm in (c) to align the enriched teacher space with the region-aware student space.

In this paper, as illustrated in Figure 2, we introduce a novel framework termed as CCKT-Det *without any additional supervision* to optimize the pipeline for efficient knowledge distillation in OVD. We argue that a cyclic and dynamic knowledge transfer from VLMs can efficiently mimic their well-aligned visual-semantic space and close the gap with methods that leverage extra data. Concretely, we first construct language queries by integrating semantic priors into the input of the detector, *i.e.*, object queries, which makes the model novel-concepts aware and accelerates model convergence (*ref* Sec. 3.2). To leverage the enriched semantics, we further propose a regional contrastive knowledge distillation loss that aligns the region-level visual features of queries from the student model with that from the teacher model (*ref* Sec. 3.3). The semantic priors and regional contrastive knowledge distillation loss together form *a cross-modal cycle with detector in between that can dynamically integrate semantic and visual knowledge from VLMs*. The cycle forces detector to tightly align with VLMs as breaking either visual or semantic alignment with VLMs would fail this cyclic knowledge transfer as evidenced by our ablation study. Unlike existing methods that enhance performance by utilizing a more complex student backbone, our method exhibits consistent performance improvements as the strength and reliability of the teacher model increase without introducing any additional computational overhead during inference. We validate the effectiveness of the proposed CCKT-Det across the COCO (Lin et al., 2014), LVIS (Gupta et al., 2019), and Objects365 (Shao et al., 2019) transfer benchmarks. On the OV-COCO (Zareian et al., 2021) benchmark, CCKT-Det enhances the $AP_{50}$ for novel categories by 2.9% compared to previous methods. This enhancement consistently scales with a stronger teacher model, culminating in a new state-of-the-art accuracy of 46.0% $AP_{50}$ on novel classes, all without the need for additional training data. On OV-LVIS (Gupta et al., 2019), CCKT-Det attains a competitive average precision of 18.2% average AP on rare without the incorporation of additional training data. Furthermore, when the model trained on the LVIS dataset is directly evaluated on the COCO and Objects365 datasets, CCKT-Det++ yields a performance of 38.5% AP and 15.2% AP, narrowing the performance difference compared to supervised models on COCO (-17%) and Objects365 (-40%), respectively.

The main contributions are summarized as the following:

- We propose semantic priors as guidance and regional contrastive knowledge distillation loss to effectively align with the visual-semantic space of VLMs.
- With the proposed two components, CCKT-Det constructs a cross-modal cycle that the detector in between can dynamically choose to transfer knowledge from.
- The performance steadily improves as the teacher VLMs and MLLMs scale, and the proposed method achieves state-of-the-art in OVD without relying on any additional training data.

## 2 RELATED WORK

**Contrastive Representation Learning.** Recent developments in self-supervised pretraining, particularly through contrastive learning, have highlighted its ability to reduce reliance on labeled datasets. These methods primarily focus on the extraction of robust visual representations on image-level tasks (He et al., 2020; Chen et al., 2020). While effective at capturing general features, such global representations face significant challenges when applied to dense perception tasks. To mitigate,

region-level contrastive learning has been introduced with different variants, such as sliding windows (Xiao et al., 2021), object proposals (Wei et al., 2021; Hénaff et al., 2021), and random query patches (Dai et al., 2021) to capture local visual clues. These strategies transform and adapt representations from image-level to pixel- or region-level, enhancing spatial reasoning capability in dense perception tasks. At the core of these methods is the definition of regional-level pre-text tasks. In our work, we leverage contrastive learning to distill rich semantic knowledge from a teacher model into the regional representations of a student model, tightly integrated with Hungarian matching to establish regional-level pairs. The proposed method captures higher-order correlations and dependencies in representation space, offering a more diverse set of negative and positive samples for distillation, thereby alleviating the constraints of learning exclusively from positive pairs of base categories.

**Vision-Language Models (VLMs) and Multimodal Large Language Models (MLLMs).** VLMs and MLLMs have profoundly transformed the integration of linguistic information in visual recognition tasks. As a prominent example of VLMs, CLIP (Radford et al., 2021), successfully aligns which image goes with which text on billion-scale image-text pairs, demonstrating strong generalization on novel categories. Concurrently, MLLMs, *e.g.*, BLIP-2 (Li et al., 2023b), MiniGPT-4 (Zhu et al., 2023), and Vicuna (Zheng et al., 2023) connect image encoders with large language models to interpret visual clues and respond to human instructions effectively. Pretraining on web-scale data has endowed VLMs and MLLMs with a remarkable capability across various downstream tasks and benchmarks (Li et al., 2023a; Liu et al., 2024; Zeng et al., 2024). In this study, we adapt VLMs and MLLMs as teachers to convey novel semantic knowledge to our proposed detector.

**Open Vocabulary Object Detection (OVD).** To transfer the rich image-level semantic knowledge in VLMs to region-level, many works (Zhu & Chen, 2024) seek to utilize additional supervisions, *e.g.*, region-word alignment on image-text pairs in a weakly supervised manner which is noisy, or pseudo annotations generated via a self-training strategy which needs to know novel categories during training. Another methodology distills from teacher VLMs to student detectors for semantic knowledge transfer. The pioneering work ViLD (Gu et al., 2021) distills region embeddings from teacher VLMs in an element-wise manner, rendering the cross-modal similarity comparison toward the well-aligned visual-semantic space of VLMs. Building upon ViLD, BARON (Wu et al., 2023a) distills from a single region to a bag-of-regions level, effectively mining the co-occurrence and compositional structure of visual concepts inherently captured by VLMs. In addition, the pretrained image encoder of VLMs can be directly leveraged as a detector backbone while adding detection heads on top of it. The backbone is either frozen (Kuo et al., 2022; Zhong et al., 2022), or fine-tuned (Kim et al., 2023b) on detection datasets. To address the distribution gap between image-level pretraining and region-level perception, miscellaneous techniques including prompt tuning (Du et al., 2022; Feng et al., 2022) have been proposed. In particular, CORA (Wu et al., 2023c) utilizes region prompting to refine CLIP image encoder for improved localization. DITO (Kim et al., 2024) introduces a shifted-window learning technique to the region-centric image-text pretraining.

## 3 METHOD

### 3.1 OVERVIEW

The objective of OVD is to train the detector on base categories $C_B$ and evaluate its performance on both base and novel categories $C_N$, where $C_B \bigcap C_N = \emptyset$ (Zareian et al., 2021; Bansal et al., 2018). As illustrated in Figure 2, our framework follows DETR-style detectors (Carion et al., 2020; Zhu et al., 2020), where learnable object queries serve as a global inductive bias that incorporate image-specific contexts. To fully exploit the regional structure embedded in the visual-semantic space of VLMs, we establish a cyclic cross-modal knowledge transfer between language-guided queries and teacher visual region features. The semantic prior embeddings are added to object queries as the input language queries to the detector decoder, while object crops are encoded by CLIP image encoder to form the teacher visual region features (Gu et al., 2021).

### 3.2 SEMANTIC PRIORS AS GUIDANCE

We first identify the problem of concept existence, *i.e.*, whether or not the concept exists in the given image. Such information acts as a semantic prior specific to the image. This problem is intrinsically dynamic and tailored to each specific image. Prefiltering out irrelevant concepts can avoid

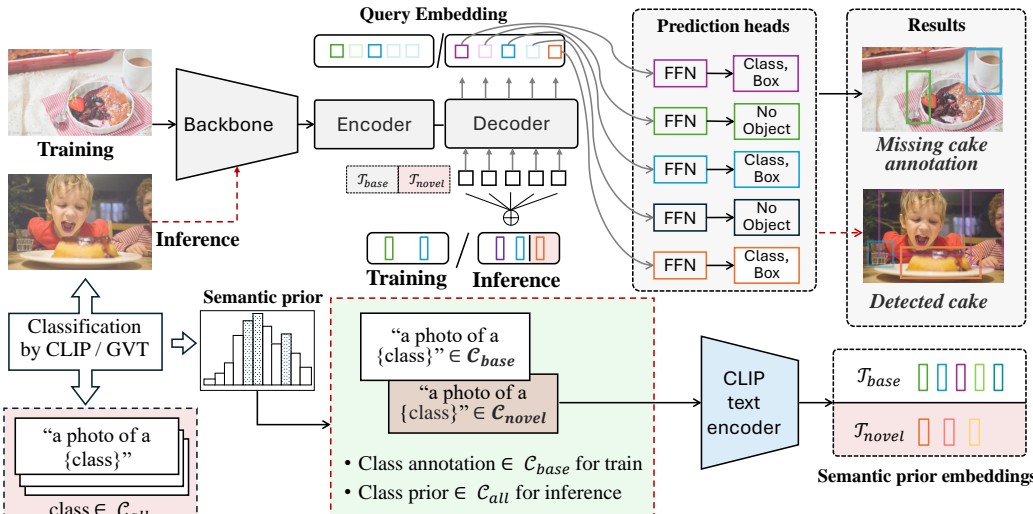

Figure 2: The overall architecture of our CCKT-Det. By querying the existence of object categories within an input image, we dynamically guide object queries to explore novel concepts using semantic priors, which enables awareness of novel categories.

the detector to falsely predict classes that are non-existent in current image and ease the burden of recognizing novel objects as the number of existent base objects is significantly reduced. However, existing methods (Zang et al., 2022; Wu et al., 2023a; Kim et al., 2024) still intervene in the prediction of existent concepts with non-existent concepts after training.

We propose two alternatives to resolve the concept existence. The first approach follows the zero-shot classification procedure of CLIP. For each image in validation set, a similarity score with text embeddings $\{t_i\} \in \mathbb{R}^{\mathcal{C}_{all} \times D}$ of all classes is computed and subsequently activated by sigmoid function. The text embeddings are encoded from CLIP text encoder by filling category names into template prompts. Logits that are below threshold $\rho$ are filtered out to ensure that only highly confident classes are identified as present ones. However, CLIP lacks fine-grained visual perception abilities and behaves like bags-of-words (Yuksekgonul et al., 2023). To bypass this drawback, we instead employ a more robust MLLM (Wang et al., 2023) that is proficient in multi-class identification to determine the presence of a specific concept within an image. For example, we prompt the MLLM with "Question: Does [OBJ] exist in the image? Answer:". The model is expected to respond with "Yes/No", making the problem of concept existence a binary classification problem. For more details of this operation, please refer to (Wang et al., 2023).

In contrast to previous methods, where object queries are static and fixed for each image after training, we dynamically inject semantic prior information into object queries to form the language queries for each given image. Specifically, the semantic prior embeddings $\{t_i\}$ are text features of those remaining classes (above the threshold $\rho$) encoded by CLIP text encoder (Radford et al., 2021) via filling their category names into template prompts. These embeddings are added to the learnable object queries to form the language queries. This process can be expressed as:

$$e = \mathcal{F}_\psi(z, q + t), \tag{1}$$

where $\mathcal{F}_\psi$ denotes the transformer decoder in detector parameterized by $\psi$, $z$ refers to visual image contexts from transformer encoder, and $q$ stands for the original learnable object queries. Note that $t$ are semantic priors tailored to each image. During training, $t$ is selected based on corresponding annotations in the given image. In inference, we first leverage the aforementioned two approaches to determine which concepts are present, then $t$ is selected according to those existent concepts. Figure 2 illustrates the forward pass of our proposed method.

**Comparisons with Conditional Matching and Language Guided Query Selection.** OV-DETR (Zang et al., 2022) uses object queries conditioned on text embeddings for class-aware regression, with pseudo annotations and self-training guiding the decoder to propose novel objects. In contrast, we incorporate the semantic priors into object queries for cyclic knowledge transfer, efficiently mining VLMs' regional structure. Our approach also avoids OV-DETR's repetitive per-class

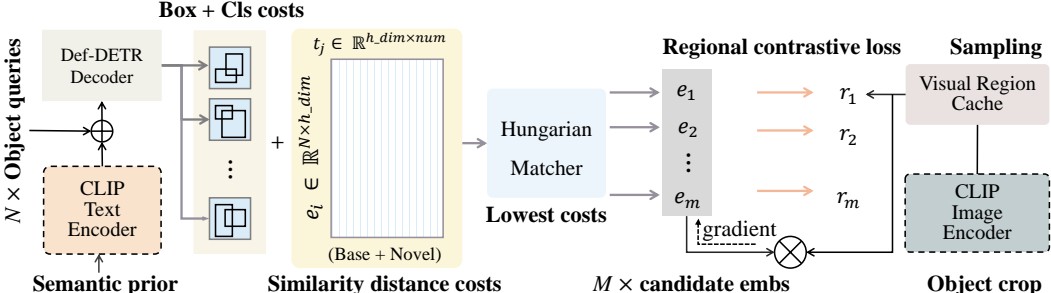

Figure 3: Illustration of our contrastive knowledge transfer scheme. We first align semantic-enriched regional embeddings with teacher's visual-semantic space through a contrastive loss. Regional embeddings with the lowest Hungarian matching cost are then considered as positive pairs for distillation, enabling explicit alignment of base objects and implicit learning of novel objects.

decoding by pre-filtering irrelevant concepts, enhancing both speed and performance. Grounding DINO (Liu et al., 2023) utilizes a text encoder to process textual inputs, selecting the top $N$ image patch features as object queries based on their maximum similarity to the text features. In contrast, our method employs teacher VLMs/MLLMs for semantic priors, which mitigates competition among similar concepts. Besides, we adapt the textual inputs to be specific to each image rather than relying on fixed inputs. While Grounding DINO's object queries may still include non-existent concepts due to cross-attention mechanisms, our approach focuses exclusively on concepts that are present in the image.

### 3.3 REGIONAL CONTRASTIVE DISTILLATION

Following previous works (Carion et al., 2020; Zhu et al., 2020), our loss function is based on the optimal bipartite matching $\hat{\sigma}$ between predicted and ground-truth objects:

$$\hat{\sigma} = \arg \min_{\sigma \in \mathfrak{S}_N} \sum_{i=1}^{N} \mathcal{L}_{match}(y_i, \hat{y}_{\sigma(i)}). \tag{2}$$

The matching cost considers the accuracy of both class prediction and bounding box regression:

$$\mathcal{L}_{match}(y_i, \hat{y}_{\sigma(i)}) = -\mathbb{1}_{\{c_i \neq \emptyset\}} \hat{p}_{\sigma(i)}(c_i) + \mathbb{1}_{\{c_i \neq \emptyset\}} \mathcal{L}_{bbox}(b_i, \hat{b}_{\sigma(i)}). \tag{3}$$

In OVD, since the detector is trained only on annotations of base classes, the prediction of detector is prone to be biased toward base categories. In contrast, VLMs already encapsulate rich semantics of novel classes in their well-aligned visual-semantic space and are superior on base-to-novel generalization. In addition to the negative log-likelihood used in original matching, we also consider probability calculated by VLMs in the matching cost, defined as:

$$\hat{p} = \sigma \left( \frac{e \cdot t^T}{\tau * ||e|| * ||t||} \right), \tag{4}$$

where $\sigma$ denotes sigmoid function, $e$ represents the visual region embeddings corresponding to object queries after the last decoder layer and projection head, and $\tau$ is temperature for sharpness scaling. In inference, we only use the probability computed by VLMs for classification.

Since $e$ in Eq. 1 contextualizes the semantic priors to the visual clues tailored to the current image as it is progressively forwarded through the decoder, the matching in Eq. 3 and Eq. 4 performs reliable similarity comparison within a single text modality. However, we argue that transferring visual knowledge from VLMs can fully recover and unleash the pretrained visual-semantic alignment. It is noteworthy that semantic priors $t$ is added to the object queries, which is the *input* to detector. At the *output* of the detector, $e$ distills visual knowledge from VLMs via our proposed regional contrastive loss, hence this cycle strictly forces the object queries to bind knowledge from both visual and text modality of VLMs. Recent text-to-image generation models (Gal et al., 2022; Ruiz et al., 2023) learn to generate personalized and subject-driven concepts by binding visual features of a concept to a randomly initialized token in the input embedding look-up table. Here, object

queries are also learnable tokens, however, in our contrastive transfer cycle, object queries learn to associate semantic priors with region-level visual features. Formally, the regional contrastive knowledge distillation loss is defined as the following:

$$\mathcal{L}_{teacher \rightarrow student} = \sum_{i=0}^{M-1} -log(\frac{\exp(r_i^T e_i/\tau)}{\sum_{j=0}^{N-1} \exp(r_i^T e_j/\tau)}), \quad (5)$$

$$\mathcal{L}_{student \rightarrow teacher} = \sum_{i=0}^{M-1} -log(\frac{\exp(e_i^T r_i/\tau)}{\sum_{j=0}^{M-1} \exp(e_i^T r_j/\tau)}). \quad (6)$$

In these equations, $M$ represents the number of ground-truth objects in each image. The Hungarian matcher selects $M$ region embeddings out of $N$ candidates from detector output to form the matching pairs as shown in Figure 3, which are then aligned with their corresponding teacher features $r$. Teacher features $r$ are extracted offline, where object crops are processed by the image encoder of teacher and stored in a cache for computational efficiency.

The overall loss function $\mathcal{L}(y, \hat{y})$ is defined as:

$$\mathcal{L}(y, \hat{y}) = \lambda_{bbox}\mathcal{L}_{bbox}(b, \hat{b}) + \lambda_{contrast}\mathcal{L}_{contrast}(r, e) + \lambda_{cls}\mathcal{L}_{cls}(p, \hat{p}), \quad (7)$$

where the bounding box loss includes both L1 and generalized IoU loss (Rezatofighi et al., 2019). The regional contrastive distillation loss $\mathcal{L}_{contrast}$ is the average of Eq. 5 and Eq. 6, and the classification loss is focal loss (Lin et al., 2017).

**Distilling from both positive and negative pairs.** The objective used in existing knowledge distillation methods for OVD is typically L1 loss that aligns the region embedding of the student model with that of the teacher model. However, the representational relation is inherently structured, with dimensions exhibiting complex inter-dependencies (Tian et al., 2019). The L1 loss is limited to distill proposals labeled as foreground ones from teacher individually. In contrast, we consider all other region candidates as negative samples, thereby providing a larger and more diverse set of negative instances for distillation. This approach can facilitate the capture of correlations and higher-order dependencies within the representation space.

## 4 EXPERIMENTS

**Datasets and evaluation metrics.** Following the OV-COCO benchmark (Zareian et al., 2021), we train the our model on 48 base categories and test on both the 48 base classes ($C_B$) and 17 novel classes ($C_N$). We also conduct experiments on LVIS (Gupta et al., 2019), where 866 common and 405 frequent classes are treated as base categories, and 337 rare classes are considered novel categories. The main evaluation metric is Average precision (AP) for novel categories, *i.e.*, $AP_{50}^N$. For OV-LVIS, as in (Wu et al., 2023a;c), we report the averaged bounding box AP across IoUs from 0.5 to 0.95 for rare classes denoted as $AP_r$.

**Implementation details.** Our model is built on Def-DETR (Carion et al., 2020). The backbone defaults to ResNet-50. We employ $3\times$ training schedule following the standard practices. Due to the markedly greater number of classes in LVIS compared to COCO, we allocate 300 queries for COCO and 600 queries for LVIS, respectively. The model is trained with semantic guidance in Sec. 3.2 for the first 30 epochs with a base learning rate $10^{-4}$. Subsequently, training continues with the incorporation of regional contrastive distillation loss, accompanied by a learning rate decay factor of 0.1 post the 31st epoch. A batch size of $2\times6$ (RTX-3090 GPUs) is employed, with AdamW optimizer. Gradient clipping is adopted with a maximum norm 0.1. Following (Carion et al., 2020; Zareian et al., 2021), $\tau$ is set to 0.07, and the loss weights are set to $\lambda_{gIoU} = 2.0$, $\lambda_{L1} = 5.0$, $\lambda_{cls} = 2.0$, and $\lambda_{contrast} = 1.0$. The number of template prompts is set to 12 following (Cho et al., 2024). For training efficiency, we follow the practice of ViLD (Gu et al., 2021) and OV-DETR (Zang et al., 2022) by extracting region crop features offline, as well as the semantic priors in Sec. 3.2.

### 4.1 MAIN RESULTS

**Comparisons with the state-of-the-arts.** Most OVD methods require weak or pseudo annotations during training. For example, BARON+ (Wu et al., 2023a) and CoDet (Ma et al., 2024) require additional caption datasets in which novel concepts can be discovered and mined. OV-DETR (Zang

| Methods | Supervisions | Backbone | $AP_{50}^N$ (%) | $AP_{50}$ (%) |
|---|---|---|---|---|
| **Annotation:** Extra caption datasets, Weak/Pseudo Labels in $C_B \cup C_N$ | | | | |
| Detic (Zhou et al., 2022) | ImageNet21K & CC3M | RN50-C4 (24M) | 27.8 | 42.0 |
| OV-DETR (Zang et al., 2022) | Pseudo annotation | RN50 (24M) | 29.4 | 52.7 |
| RegionCLIP (Zhong et al., 2022) | CC3M & COCO Caption | RN50×4 (87M) | 39.3 | 55.7 |
| CoDet (Ma et al., 2024) | CC3M & COCO Caption | RN50 (24M) | 30.6 | 46.4 |
| BARON+ (Wu et al., 2023a) | COCO Caption | RN50-C4 (24M) | 42.7 | 51.7 |
| CORA+ (Wu et al., 2023c) | COCO Caption | RN50×4 (87M) | 43.1 | 56.2 |
| CFM-ViT (Kim et al., 2023a) | LAION-2B | ViT-L/16(307M) | 34.3 | 46.4 |
| DITO (Kim et al., 2024) | LAION-2B | ViT-B/16 (86M) | 36.6 | 48.8 |
| DITO (Kim et al., 2024) | DataComp-1B | ViT-L/16(307M) | 40.2 | 54.6 |
| **Annotation:** Instance-level labels in $C_B$ | | | | |
| ViLD-ens (Gu et al., 2021) | CLIP | RN50-FPN (24M) | 27.6 | 51.3 |
| F-VLM (Kuo et al., 2022) | CLIP | RN50-FPN (24M) | 28.0 | 39.6 |
| BARON (Wu et al., 2023a) | CLIP | RN50-FPN (24M) | 34.0 | 53.5 |
| CORA (Wu et al., 2023c) | CLIP | RN50 (24M) | 35.1 | 35.4 |
| CORA+ (Wu et al., 2023c) | CLIP | RN50×4 (87M) | 41.7 | 43.8 |
| BIND (Zhang et al., 2024) | CLIP | ViT-B/16 (86M) | 36.3 | 50.2 |
| BIND (Zhang et al., 2024) | CLIP | ViT-L/16 (307M) | 41.5 | 54.8 |
| CLIP-Self (Wu et al., 2023b) | CLIP | ViT-B/16 (86M) | 37.6 | - |
| CLIP-Self (Wu et al., 2023b) | CLIP | ViT-L/14 (307M) | 44.3 | - |
| OV-DQUO (Wang et al., 2024) | CLIP | RN50×4 (87M) | 45.6 | - |
| CCKT-Det (ours) | CLIP | RN50 (24M) | **38.0** | 35.0 |
| CCKT-Det (ours) | CLIP | SwinB (88M) | **41.9** | 40.9 |
| CCKT-Det++ (ours) | CLIP | RN50 (24M) | **45.3** | 46.9 |
| CCKT-Det++ (ours) | CLIP | SwinB (88M) | **46.0** | 46.2 |

Table 1: Result comparisons on OV-COCO (Zareian et al., 2021). Methods are categorized into two groups based on whether additional supervisions beyond instance-level labels in $C_B$ are utilized during training *e.g.*, extra image-text datasets, weak or pseudo labels. CCKT-Det++ denotes that we use more reliable semantic priors filtered by GVT (Wang et al., 2023) in inference.

| Methods | Supervisions | Backbone | $AP_r$ (%) | $AP$ (%) |
|---|---|---|---|---|
| ViLD (Gu et al., 2021) | CLIP | RN50-FPN (24M) | 16.3[bbox] | 24.4[bbox] |
| OV-DETR (Zang et al., 2022) | CLIP & Pseudo annotation | RN50 (24M) | 17.4[mask] | 26.6[mask] |
| CCKT-Det++ (ours) | CLIP | RN50 (24M) | 18.2[bbox] | 27.1[bbox] |
| RegionCLIP (Zhong et al., 2022) | CLIP & COCO Caption | R50×4 (87M) | 22.0[mask] | 32.3[mask] |
| CORA+ (Wu et al., 2023c) | CLIP & COCO Caption | RN50×4 (87M) | 28.1[bbox] | - |
| CoDet (Ma et al., 2024) | COCO Caption & CC3M | SwinB (88M) | 29.4[mask] | 39.2[mask] |
| CCKT-Det++ (ours) | CLIP & Pseudo annotation | SwinB (88M) | 32.8[bbox] | 44.3[bbox] |

Table 2: Result comparisons on OV-LVIS (Gupta et al., 2019). CCKT-Det++ achieves competitive performance with ResNet-50 (*i.e.*, RN50) as backbone and without any extra data. Performance would further boost when larger backbone and pseudo annotations available.

et al., 2022) goes a step further by generating pseudo annotations in a self-training manner. Table 1 presents our results on OV-COCO (Zareian et al., 2021). Among methods that utilize additional supervision, CORA+ using COCO caption dataset achieves the best with $AP_{50}^N$ 43.1% on novel classes, with a comparable backbone SwinB, our CCKT-Det closely matches CORA+ ($AP_{50}^N$ 41.9%) without any additional data during detector training stage, demonstrating the superior data efficiency of our method. When equipped with a stronger teacher model GVT (Wang et al., 2023) for more reliable semantic guidance as in Sec. 3.2, our CCKT-Det++ achieves a new state-of-the-art performance with 46.0% $AP_{50}^N$ compared to all methods. We also present result comparisons with the state-of-the-arts on OV-LVIS in Table 2. Initially, we train CCKT-Det on the LVIS dataset without using additional data, achieving competitive performance compared to ViLD (Gu et al., 2021) and OV-DETR (Zang et al., 2022). However, it falls behind state-of-the-art methods that leverage caption supervision.

| Methods | COCO (Lin et al., 2014) | | | Objects365 (Shao et al., 2019) | | |
|---|---|---|---|---|---|---|
| | AP (%) | $AP_{50}$ (%) | $AP_{75}$ (%) | AP (%) | $AP_{50}$ (%) | $AP_{75}$ (%) |
| Supervised (Gu et al., 2021) | 46.5 | 67.6 | 50.9 | 25.6 | 38.6 | 28.0 |
| ViLD (Gu et al., 2021) | 36.6 | 55.6 | 39.8 | 11.8 | 18.2 | 12.6 |
| DetPro (Du et al., 2022) | 34.9 | 53.8 | 37.4 | 12.1 | 18.8 | 12.9 |
| F-VLM (Kuo et al., 2022) | 32.5 | 53.1 | 34.6 | 11.9 | 19.2 | 12.6 |
| BARON (Wu et al., 2023a) | 36.2 | 55.7 | 39.1 | 13.6 | 21.0 | 14.5 |
| CCKT-Det++ (ours) | 38.5 | 53.2 | 42.1 | 15.2 | 20.9 | 15.8 |

Table 3: Result comparisons of the LVIS-trained model evaluated on the COCO (Zareian et al., 2021) and Objects365 (Shao et al., 2019) datasets are presented without any fine-tuning.

| # | Def-DETR | Semantic Guiding | Regional-Contrastive Loss | Similarity Classification | $AP_{50}^N$ (%) |
|---|---|---|---|---|---|
| 1 | ✓ | ✗ | ✗ | ✗ | 25.4 |
| 2 | ✓ | ✓ | ✗ | ✗ | 32.6 |
| 3 | ✓ | ✓ | ✓ | ✗ | 31.7 |
| 4 | ✓ | ✗ | ✓ | ✓ | 33.7 |
| 5 | ✓ | ✓ | ✓ | ✓ | **38.0** |

Table 4: The ablation study results by integrating different components of our method, including concerning semantic guidance, regional contrastive loss, and similarity classification (*i.e.*, a facilitating operation in the post-processing stage). While semantic guidance demonstrates effectiveness to a degree, its efficacy is constrained in the absence of regional contrastive training. Furthermore, to fully harness the capabilities of the well-trained model, it is essential to employ it alongside the similarity-based classification post-processing. The comprehensive integration of all components effectively completes the operational framework of our CCKT-Det.

This is attributed to our use of naive hand-crafted prompts for semantic priors as guidance, which may lack robustness in distinguishing fine-grained concepts in the LVIS label space. To address this, we enhance the dataset by incorporating highly confident (*i.e.*, 0.9) predictions from CCKT-Det into the base categories and train a more robust version, which results in 32.8% AP on rare classes.

**Results on transfer detection setting.** Table 3 gives our results under the cross-dataset transfer evaluation setting. We train on OV-LVIS and evaluate on COCO and Objects365 v1 (Shao et al., 2019) datasets without any additional fine-tuning. We can observe that our method demonstrates superior performance, surpassing F-VLM by +6.0%/+3.3% AP and BARON by +2.3%/+1.6% AP, while significantly reducing the performance gap compared to supervised models on COCO (-17%) and Objects365 (-40%). CCKT-Det++ shows better performance especially at a higher IoU threshold of $AP_{75}$, demonstrating superior accuracy and reliability compared to its counterparts.

## 4.2 ABLATION ANALYSIS

We conduct ablation studies on OV-COCO (Zareian et al., 2021) to evaluate the effectiveness of each proposed component, hyperparameters, and scalability. Unless stated otherwise, experiments were performed using a ResNet50 backbone and a $3\times$ training schedule.

**Semantic priors as guidance.** Table 4 ablates on our proposed two components. We can observe that after removing the guidance of semantic priors, the language queries revert to the default object queries in Def-DETR, it leads to a significant performance drop from 32.6 AP to 25.4 AP, highlighting the importance of semantic guidance in enhancing the model ability to detect novel objects. Relying solely on semantic priors as guidance, we achieve 32.6 AP on novel classes even without any extra supervision, surpassing OV-DETR and CoDet that exploit additional data.

**Regional contrastive distillation.** As validated by the difference between the row 2 and row 5 in Table 4, the absence of distillation loss results in a performance decline of 5.4% $AP_{50}$ on novel classes. As elaborated in Sec. 3.3, our regional contrastive loss establishes a cyclical relationship with semantic priors, thereby fully leveraging the visual-semantic space of VLMs. *Therefore, we can obtain the conclusion that the regional contrastive loss should be used in conjunction with similarity classification to form cyclic knowledge transfer loop.* Row 3 and row 5 in Table 4 are models with identical weights, the only difference lies in how they give classification predictions

| # | Teacher | Backbone | $AP_{50}^N$ (%) | Epochs |
|---|---------|----------|-----------------|--------|
| OV-DETR | ViT-B-32 | ResNet50 | 29.4 | 50 |
| 1 | ViT-B-16 | ResNet50 | 35.2 | 36 |
| 2 | ViT-L-14 | ResNet50 | 38.0 | 36 |
| 3 | ViT-H-14 | SwinB | 41.9 | 12 |

Table 5: Performance grows steadily as the teacher becomes stronger, while maintaining the backbone parameters at a moderate level.

| Semantic Priors | Threshold $\rho$ | $AP_{50}^N$ (%) |
|-----------------|------------------|-----------------|
| 6 | 0.70 | 38.0 |
| 6 | 0.90 | 37.6 |
| 12 | 0.70 | 36.8 |
| 12 | 0.90 | 36.2 |

Table 6: Ablation study results on semantic priors. Performance is influenced by the accuracy of classification ability of teacher.

in inference, and row 5 is a similarity-score classification while line 3 derives the prediction that incorporates logits score from classification branch. Since the same model presents disparate results with different post-processes, we reckon the reason behind is: after training, the embedding space of CCKT-Det is pulled close to the hidden space of the teacher. Specifically, added with semantic prior features, the queries are refined by the decoder, then guided to be aligned with visual features encoded by the teacher. The integration of textual and visual features within the detection loop sets the optimization direction towards mimicking teacher on region level. Using vanilla post-process to give classification score fails to fully unleash the model's full potential. Instead, the classification predictions should align with the classification methodology of the teacher. For instance, CLIP (Radford et al., 2021) generates zero-shot class predictions based on similarity scores. These holistic operations collectively complete the cycle of our cyclic knowledge transfer for OVD.

**Scaling to stronger teacher models.** To evaluate the effectiveness of our knowledge transfer strategy, we keep the backbone parameters at a moderate level while scaling with stronger teacher models. As shown in Table 5 and Figure 4, our method can consistently improve with more robust teacher models. This scaling effect is manifested in two ways: 1) teachers trained on larger datasets with more parameters have a more aligned visual-semantic space from which our regional contrastive loss can transfer to the detector; 2) in the inference phase, we employ more robust teachers to provide more accurate semantic priors that dynamically transfer semantic knowledge to the detector. Notably, other approaches (Wu et al., 2023c; Kuo

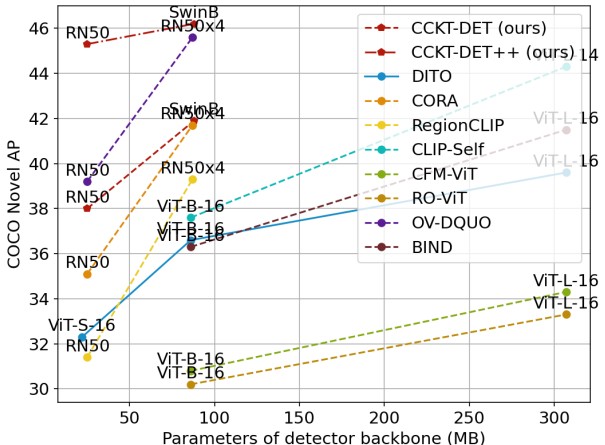

Figure 4: While most models tend to enhance performance with stronger backbones, typically resulting in increased computational demands, our method achieves competitive results utilizing the default ResNet50 backbone.

et al., 2022) often directly scale the backbone which is the teacher model itself, resulting in a substantial computational burden (about 3 × more training time), as evidenced by models like ResNet50x4 (87M parameters) to ResNet50x64 (420M parameters) used in F-VLM (Kuo et al., 2022), ViT-L/14 (307M parameters) and EVA02-L (304M parameters) in CoDet (Ma et al., 2024). Our method keeps student with a moderate size but achieves comparative performance by distilling from stronger teachers and more reliable prior information. Our method also shows faster convergence compared to the standard setting used in Def-DETR (Zhu et al., 2020) and OV-DETR (Zang et al., 2022).

**Number of semantic priors.** As discussed in Sec. 3.2, during inference, we leverage CLIP to resolve the concept existence problem, compensating for the absence of prior information available during training for each image. Specifically, we introduce a hyperparameter $L$ to represent the maximum number of concepts an image may contain. Concepts are identified only if their scores exceed a threshold $\rho$. If the number of present concepts identified is less than $L$, the remaining concepts are randomly sampled from $L$ until $L$ is met. However, this process may introduce noisy

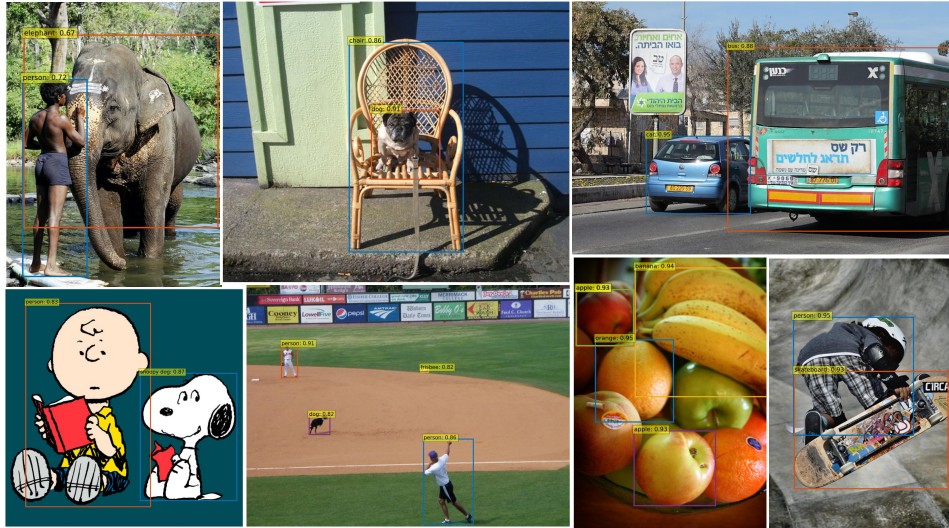

Figure 5: More visualization results of CCKT-Det++.

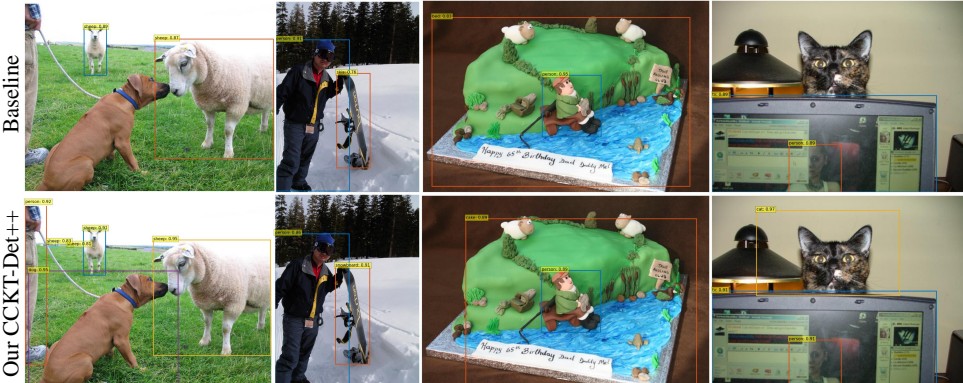

Figure 6: Visualization results of CCKT-Det++ compared with the baseline model.

concepts into the model as shown in Table 6. This motivates us to resort to stronger MLLMs that are good at multi-class identification like GVT (Wang et al., 2023) to provide more reliable prior.

### 4.3 VISUALIZATION RESULTS

In Figure 5, we visualize more prediction results of CCKT-Det++ on images with novel objects. It is evident that our method achieves accurate object detection results, with all these images encompassing both base categories and novel categories. In Figure 6, we present object detection results of CCKT-Det++ alongside a baseline model Def-DETR (Zhu et al., 2020) (*i.e.*, corresponds to line 1 in Table 4). In the first line and the last line, CCKT-Det++ correctly detects novel objects "*dog*" and "*cat*". In the second line and the third line, the baseline model incorrectly labels "*skis*" as "*snowboard*" and "*bed*" as "*cake*", while CCKT-Det++ correctly detected those novel objects.

## 5 CONCLUSION

We propose to leverage semantic priors as guidance and regional contrastive knowledge distillation loss that cyclicly and dynamically transfer knowledge from both text and image encoder of VLMs. Unlike previous state-of-the-arts that use extra supervisions, we exploit the regional structure embedded in the well-aligned visual-semantic space of VLMs without any additional data. The performance scales consistently as the teacher model is stronger while keeping the backbone parameters at a moderate level. We hope our findings could inspire the community to further explore the hidden space of VLMs and MLLMs for downstream perception tasks. In the future, we will explore more efficient and lightweight approaches for open-vocabulary object detection.

ACKNOWLEDGMENT

The authors would like to thank all anonymous reviewers for their positive comments and constructive suggestions. This work was partially supported by the Hong Kong SAR RGC General Research Fund under Grant 16208823 and ACCESS - AI Chip Center for Emerging Smart Systems, sponsored by InnoHK funding, Hong Kong SAR. Long Chen was supported by the Hong Kong SAR RGC Early Career Scheme (26208924), the National Natural Science Foundation of China Young Scholar Fund (62402408), and the HKUST Sports Science and Technology Research Grant (SSTRG24EG04).

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

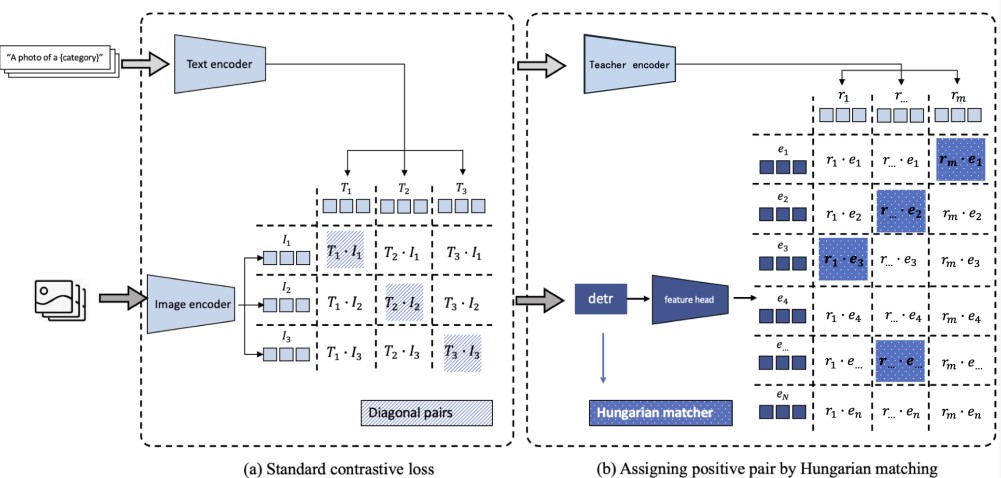

Figure 7: Illustration of how to form positive pairs.

## A DETAILS ABOUT IMPLEMENTING CONTRASTIVE LOSS

In the implementation of contrastive loss, the initial step involves the construction of a score matrix that quantifies the similarity between the representations of two instances within an embedding space. Following this, the info-NCE loss is computed, where the numerator consists of the similarity scores derived from positive pairs. The denominator is subsequently calculated by aggregating the exponentiated similarity scores across all pairs, which includes both positive and negative instances. The loss function is defined as the negative log likelihood of the positive pair similarities relative to the total similarities across all pairs. This methodology promotes the maximization of similarities for positive pairs while concurrently minimizing those for negative pairs, thereby effectively organizing the structure of the embedding space.

In most contrastive learning frameworks, positive pairs are typically identified by the diagonal elements of the similarity matrix, as these elements reflect instances compared to themselves, representing the highest similarity within a batch. However, this paradigm does not align with our approach. In our context, regional-level object candidates do not inherently correspond to instance-level visual features from teacher, thereby rendering the diagonal elements of the score matrix ineffectual. As previously discussed, we establish positive pairs through Hungarian matching. Consequently, from an implementation standpoint, we delineate positive pairs based on the outcomes of Hungarian matching within the asymmetric similarity matrix.

