# OpenReview forum: "Cyclic Contrastive Knowledge Transfer for Open-Vocabulary Object Detection"
_ICLR.cc/2025/Conference — ICLR 2025 Poster_

### Official Review · Reviewer_TQtb · 2024-10-28

**Soundness:** 2
**Presentation:** 3
**Contribution:** 3
**Rating:** 6
**Confidence:** 5

**Summary:**

The paper introduces Cyclic Contrastive Knowledge Transfer for Open-Vocabulary Object Detection (CCKT-Det), a method designed to enhance the detection of novel objects in open-vocabulary settings without relying on additional supervision like image-text pairs or pseudo annotations. It leverages pretrained vision-language models (VLMs) to transfer knowledge effectively from base categories to novel ones. The key contributions include using semantic priors to guide object queries and a regional contrastive knowledge distillation mechanism to align the visual features between a student and teacher model. CCKT-Det achieves state-of-the-art performance on COCO and LVIS benchmarks, improving novel object detection without extra data, and scales efficiently with stronger VLMs while maintaining moderate computational overhead.

**Strengths:**

The paper is well-structured, with solid methodological rigor. The authors present comprehensive experiments and ablation studies that demonstrate the effectiveness of their proposed method. The proposed CCKT-Det framework has high significance in advancing the state of open-vocabulary object detection. By eliminating the need for additional supervision and showing competitive performance with existing methods that do require extra data, the paper addresses a key limitation in the field.

**Weaknesses:**

- Although you claim and emphasized CCKT-Det did not rely on the extra data, in contrary to the previous method, you use the extra MLLM to be a discriminator and generate the prior semantic guidance.

**Questions:**

- In the paper, you mention, “In contrast to previous works, where object queries are static and fixed for each image after training, we dynamically inject text embeddings {ti} as semantic priors into object queries to form the language queries for each image.” However, similar ideas have been explored in earlier works, such as GroundingDINO. It would be appropriate to supplement your references by citing these related works and discuss their similarities and differences.
- In Table 4, the paper states that “While semantic guidance demonstrates effectiveness to a degree, its efficacy is constrained in the absence of regional contrastive training.” However, contrary to this claim, regional contrastive appears to have a detrimental effect on semantic guidance, as the ablation experiment did not demonstrate any significant effectiveness of regional contrastive training. This seems to contradict the statement in the paper, “As indicated in the third row of Table 4, the absence of this loss results in a performance decline of 5.4% AP50 on novel classes.” Based on the results, there appears to be no significant relationship between Regional-Contrastive Loss and Semantic Guiding, while it works in conjunction with Similarity Classification. This seems to be a clerical error.

---

> ### Author Response · Authors · 2024-11-22
> **Response to Reviewer TQtb (part1)**
>
> Thank you for your insightful feedback and for highlighting the strengths of our work, including the structured methodology, comprehensive experiments, and the significance of the CCKT-Det framework in advancing OVD. We appreciate your recognition of our approach to addressing key limitations without requiring additional supervision. The following is our response with necessary explnations.
>
> **Q1.** Similarities and differences with GroundingDINO.
>
> **Reply:** Thanks for your suggetion, we add more descriptions in refined paper to make the comparison more comprehensive following your advice.
>
> Concretely, though Grounding DINO [R10] bears certain similarity with our proposed CCKT-Det in spirit at first glance, the following aspects differentiates ours from theirs:
>
> - **The formulation of semantic priors:** For each incoming image, Grounding DINO forwards the same texts into text encoder usually extracted from dataset labels. Then top *N* pixel features from image backbone are selected and treated as object queries based on their max similarity scores with text features. However, our proposed mechanism for selecting semantic priors are based on MLLMs, which is indeed a binary classification for concept existence problem. This eliminates the potential interventions among similar concepts since similar concepts may compete with each other thus the similarity score might fall out of the top *N* operation. More importantly, we forward different texts tailored to each input image rather than statically fixed input texts. The object queries in Grounding DINO may still integrate text features of non-existent concepts in current image from its text cross-attention in cross-modality decoder as in Fig. 3 of their paper. In contrast, our object queries only learn from text features of existent concepts.
> - **Different architectures:** Grounding DINO injects additional cross-modality fusion after text encoder and image backbone, which is proved to improve performance [R11], while we follow the standard Deformable DETR architecture with no post cross-modality fusion.
> - **Different training data:** Grounding DINO uses various source pre-training data (object 365, GoldG, COCO, RefCOCO, etc.) while we keep a modest data usage to demonstrate the effectiveness of CCKT-Det.
>
> References:
>
> [R10] Grounding DINO: Marrying DINO with Grounded Pre-Training for Open-Set Object Detection. In ECCV 2024.
>
> [R11] TransVG++: End-to-End Visual Grounding with Language Conditioned Vision Transformer. In TPAMI 2022.

---

> ### Author Response · Authors · 2024-11-22
> **Response to Reviewer TQtb (part2)**
>
> **Q2.** The relationship between regional-contrastive distillation loss and semantic guidance.
>
> **Reply:** Thank you for pointing out that
> > “As indicated in the third row of Table 4, the absence of this loss results in a performance decline of 5.4% AP50 on novel classes.”
>
> It is indeed a clerical error. We are sorry for the mistake and misunderstanding inccurred. The correct version should be the following _"As indicated by the difference between the first row and last row of Table 4, the absence of this loss results in a performance decline of 5.4% AP50 on novel classes."_
>
> **Effectiveness of regional contrastive training:** It's worth noting that the results of row 2 and row 4 comes from the same model weights. The only difference between these two is how the model gives class prediction: whether using the similarity-based classification or not. In other words, the abaltion experiment comes from three different parameters that learned with three different strategies.
>
> |  Model version | Semantic Guiding | Regional-Contrastive Distillation | $AP_{50}^{N}$ (%) |
> |----------------|------------------|-----------------------------------|-------------------|
> | 1              | ✓                | ✗                                 | 32.6              |
> | 3              | ✗                | ✓                                 | 33.7              |
> | 4              | ✓                | ✓                 	            | 38.0              |
>
> These three models corresponds to the version whether trained using the components (Semantic Guiding, Regional-Contrastive Distillation) introduced in Sec. 3.
>
> In our preliminary experiment, we observed the performance of version 1 stagnates even with longer training schedule. **After incorporating the regional contrastive loss, the performace boosts from 32.6 to 38.0.** This phenomenon also indicates regional contrastive distillation also helps model get out of local optima to some exent.
>
> |  #   | Semantic Guiding | Regional-Contrastive Loss | Similarity Classification | $AP_{50}^{N}$ (%) |
> |------|------------------|---------------------------|---------------------------|-------------------|
> | 0    | ✗                | ✗                         | ✗                         | 25.4              |
> | 1    | ✓                | ✗                         | ✗                         | 32.6              |
> | 2    | ✓                | ✓                         | ✗    		              | 31.7              |
> | 3    | ✗                | ✓                         | ✓    		              | 33.7              |
> | 4    | ✓                | ✓                         | ✓    		              | 38.0              |
>
> **Regional Contrastive loss should be used in conjunction with Similarity Classification to form cyclic knowledge transfer loop**. Since the same model presents disparate results with different post-process, we reckon the reason behind is: after training, the embedding space of our detecor model is pulled close to the hidden space of the teacher. Spcifically, added with textual features, the regional-level query embeddings are refined by decoder, then guided to be aligned with visual features encoded by the teacher. The tight integration of textual and visual features within the detection loop set the optimization direction towards the student detector imitating its teachers' in a regional level.
>
> Using vanilla post-process to give classification score fails to fully unleash the model's ability. It should be used in conjuction with its teacher's classification style--CLIP also give zero-shot class predictions by similarity score. the holistic operations complete the loop of our cyclic knowledge transfer for OVD.

---

> > ### Comment · Reviewer_TQtb · 2024-11-26
> >
> > Thank you for the authors' detailed reply. My concerns have been thoroughly addressed, and I have no further questions. I have decided to maintain my current rating.

---

> > > ### Author Response · Authors · 2024-11-27
> > > **Many thanks for your confirmation**
> > >
> > > Thank you for your support of our work and for the diligence you showed in the review process.

---

### Official Review · Reviewer_kmme · 2024-11-01

**Soundness:** 3
**Presentation:** 3
**Contribution:** 3
**Rating:** 5
**Confidence:** 3

**Summary:**

The paper introduces CCKT-Det, a novel framework for open-vocabulary object detection (OVD) that operates without any additional supervision beyond the base training set. The method leverages semantic priors and regional contrastive knowledge distillation to align the detector with the visual-semantic space of vision-language models (VLMs).  This paper presents semantic priors to boost the detector's capacity for novel object recognition and a unique loss function that heightens the detector's sensitivity to new objects by aligning region-level features. Based on these contributions, this work achieves start-of-the-art results on COCO benchmark.

**Strengths:**

* The paper introduces a novel framework, CCKT-Det, for open-vocabulary object detection (OVD) that does not rely on any additional supervision beyond the base training set and achieves start-of-the-results on the COCO benchmark.
﻿
* Comprehensive experiments are conducted on multiple benchmarks (OV-COCO, LVIS, Objects365), demonstrating the method's effectiveness. The results show consistent performance improvements as the strength of the teacher model increases.  The paper includes thorough ablation studies to validate the effectiveness of each component of the proposed method.

* The paper demonstrates that high-performance OVD can be achieved without the need for additional supervision, such as image-text pairs or pseudo annotations. This is a significant advancement, especially in scenarios where such data may not be readily available.

**Weaknesses:**

* It is better to add a more detailed result comparison in Tab.1 and Tab.2. For example, show the AP_Base result for COCO and AP_c and AP_f results for LVIS.
* This method has shown promising results in detecting novel objects, but it seems not very good for detecting base objects.

**Questions:**

None

---

> ### Author Response · Authors · 2024-11-22
> **Response to Reviewer kmme (part 1)**
>
> We appreciate your acknowledgment of our method as a novel framework for the OVD task, recognizing it as a significant advancement, particularly in scenarios where data may not be readily available. Here's our response with some necessary explanations.
>
> **Question:** Report performance on base categories on COCO and LVIS.
>
> > It is better to add a more detailed result comparison in Tab.1 and Tab.2. For example, show the AP_Base result for COCO and AP_c and AP_f results for LVIS.
>
> > This method has shown promising results in detecting novel objects, but it seems not very good for detecting base objects.
>
> **Reply:** Thanks for your advice, here we report results including performance of base classes on COCO and LVIS dataset.
> Results on the COCO dataset:
>
> | Model      |   $\text{AP}_{50}^{N}$ (%) | $\text{AP}_{50}^{B}$ (%) | $\text{AP}_{50}$ (%) | $\triangle \text{AP}_{50}$ (%) $\downarrow$|
> |------------|----------------------------|--------------------------|----------------------|-------------------------------|
> |ViLD-img    |    24.1    		          | 34.2                     | 31.6                 |10.1                           |
> |F-VLM       |    28.0    		          | -                        | 39.6                 |-                              |
> |OV-DETR     |    29.4    		          | 61.0                     | 52.7                 |31.6                           |
> |CORA        |    35.1    		          | 35.5                     | 35.4                 |0.4                            |
> |CoDet       |    30.6    		          | 52.3                     | 46.6                 |21.7                           |
> |CCKT-Det    |    38.0    		          | 34.1                     | 35.0                 |3.9                            |
> |CCKT-Det (SwinB)| 41.9    		          | 40.6                     | 40.9                 |1.3                            |
> |CCKT-Det++ (SwinB)    |   46.0           | 46.3                     | 46.2                 |0.3                            |
>
> We note that on base classes, our method is not as good as others. However, previous approaches might suffer from issue of overfitting on base classes. Here we define a new metric termed $\triangle \text{AP}_ {50}=|\text{AP}_ {50}^{B}-\text{AP}_ {50}^{N}|$ to quantitatively measure this phenomenon. The $\triangle \text{AP}_ {50}$ reflects the gap between performances on base and novel classes, open-vocabulary detection should achieve both high $\text{AP}_ {50}^{B}$ and $\text{AP}_ {50}^{N}$, but more importantly, a low $\triangle \text{AP}_ {50}$ to avoid skewed performance. As this new metric shows in the above table, the overfitting phenomenon is rampant especially on methods relying on extra image-caption pairs (CoDet) and self-training techniques (OV-DETR) since the generated pseudo boxes of base classes are more accurate than pseudo boxes of novel classes. Though our method is relatively low on base classes, we maintain a good balance between base and novel classes, achieving the lowest 0.3 $\triangle \text{AP}_ {50}$.

---

> ### Author Response · Authors · 2024-11-22
> **Response to Reviewer kmme (part2)**
>
> We also conduct a new experiment that trains CCKT-Det using data provided by OV-DETR (it includes generated pseudo boxes) to support our guess: we can observe performance of base objects surpassing novel objects by integrating more generated pseudo boxes as supervisions
>
> | Model       |$AP_{50}^{N}$ (%) | $AP_{50}^{B}$ (%) | $AP_{50}$ (%)     |$\triangle \text{AP}_{50}$ (%) $\downarrow$|
> |-------------|------------------|-------------------|-------------------|-------------------------------|
> |OV-DETR      |    29.4    		 | 61.0              | 52.7              |31.6                           |
> |CCKT-Det     |    38.0    		 | 34.1              | 35.0              |3.9                            |
> |CCKT-Det*    |   47.9 		     |  51.3             |    48.6           |3.4                            |
>
> If an open-vocabulary detector can only achieve superior performance on base categories but poor performance on novel classes, its effectiveness will be diminished, given that the evaluation of OVD essentially reflects not only AP but also the model's generalization ability to novel categories.
>
> Since the hidden space of VLMs exhibits minimal bias regardless of the division of base and novel classes on COCO dataset under OVD setting [R4], there is no distinct 'base' or 'novel' for VLMs, this balanced structural knowledge is effectively distilled to our detector. A similar trend can also be observed in other knowledge transfer or knowledge distillation-based methods, such as ViLD-img and CORA, achiving a relatively low $\triangle \text{AP}_ {50}$ compared to methods that utilize additional training data.
>
>
> In other words, the student performance is affected by teacher. If teacher has limitations, they could propagate to student. We can further observe that when we employ a more robust MLLM (GVT) to guide our model, the performance on both base and novel classes is boosted (c.f. to CCKT-Det and CCKT-Det++ in Tab. 1).
>
> To summarize, there is an inherent trade-off between base and novel categories, knowledge distillation based method could inherit both balanced semantics and limitations from the teacher. With the major evaluation metric for open-vocabulary detection being the novel AP, our method achieves superior performance while striking a good balance between base and novel classes.
>
> Results on LVIS dataset with $\text{AP}_ {c}$ and $\text{AP}_ {f}$:
>
> | Model      |   $AP_{r}$ (%) | $AP_{c}$ (%) | $AP_{f}$ (%) | $AP$ (%)       |
> |------------|----------------|--------------|--------------|----------------|
> |ViLD        |$16.3^{bbox}$   |$21.2^{bbox}$ |$31.6^{bbox}$ |$24.4^{bbox}$   |
> |OV-DETR     |$17.4^{mask}$   |$25.0^{mask}$ |$32.5^{mask}$ |$26.6^{mask}$   |
> |F-VLM       |$18.6^{mask}$   |-             |-             |$24.2$          |
> |CORA+       |$28.1^{bbox}$   |-             |-             |-               |
> |CoDet       |$29.4^{mask}$   |$39.5^{mask}$ | $43.0^{mask}$|$39.2^{mask}$   |
> |CoDet (EVA02-L-304M)  |$37.0^{mask}$ |$46.3^{mask}$ |$46.3^{mask}$ |$44.7^{mask}$ |
> |CCKT-Det++  |$32.8^{bbox}$   |$45.3^{bbox}$ |$45.5^{bbox}$ |$44.3^{bbox}$   |
>
> [R4] Open-Vocabulary Object Detection Using Captions. CVPR 2021.

---

> ### Comment · Area_Chair_3RPh · 2024-11-27
>
> Dear reviewer,
>
> Today is the last day for reviewers to ask questions to authors. Did the authors' rebuttal address your concern? Do you have any additional questions?

---

> ### Comment · Reviewer_kmme · 2024-11-27
>
> I decide to maintain my original rating after reading the other reviewers' questions and your reply.

---

> > ### Author Response · Authors · 2024-12-02
> > **Response to Reviewer kmme (part4)**
> >
> > Thank you for your thoughtful evaluation of our paper and for recognizing the strengths of our proposed framework. We appreciate your feedback regarding the detailed result comparisons and the performance of base object detection.
> >
> > While we understand your decision to maintain your original rating, we hope you might consider whether our responses and the clarifications provided could lead to a more favorable assessment. We believe that addressing your concerns could enhance the overall understanding of our contributions.
> >
> > If there are any additional questions or areas of uncertainty that we can clarify further, please do not hesitate to let us know. We are eager to improve our work based on your valuable insights. Thank you once again for your time and consideration.

---

> ### Author Response · Authors · 2024-11-27
> **Response to Reviewer kmme (part3)**
>
> Thanks for your response and for taking the time to consider our rebuttal. Based on your suggestions, we have conducted the relevant experiments and provided an analysis of the results. Besides, we have addressed the second concern mentioned in the "Weaknesses" section. If you have any further comments, suggestions, or points that require clarification, please let us know. Thank you very much!

---

### Official Review · Reviewer_s2q1 · 2024-11-01

**Soundness:** 3
**Presentation:** 2
**Contribution:** 3
**Rating:** 6
**Confidence:** 3

**Summary:**

This paper presents CCKT-Det, a method for open-vocabulary object detection (OVD) that eliminates the need for extra supervision like image-text pairs or pseudo-labels. It leverages cyclic and dynamic knowledge transfer between language queries and visual region features from pretrained vision-language models (VLMs). The approach uses semantic priors for guiding query learning and a regional contrastive loss to enhance detection of novel objects. CCKT-Det achieves significant performance gains on the COCO benchmark, showing scalability with stronger VLMs while maintaining moderate computational requirements.

**Strengths:**

1. The proposed method is quite reasonable and can effectively transfer the visual and language capabilities of VLMs.
2. The experiments on the OV-COCO benchmark are relatively comprehensive, showing a significant improvement in detecting novel categories.
3. The ablation experiments are relatively thorough and comprehensive.

**Weaknesses:**

1. The introduction to the task setup is limited, and there is a lack of a focused, clear description and visualization of the workflow details of CCKT-Det during training and testing. Figure 2 is somewhat confusing.
2. The AP performance of CCKT-Det on the OV-COCO benchmark is relatively poor, and its detection capability on base categories is not as strong as other methods. This may be the cost of improving detection capabilities for novel categories.
3. The experimental results did not report the AP for the base category.
4. There is no comparison or analysis of time complexity.

**Questions:**

1. Why does the Regional-Contrastive Loss in the ablation study cause drop of AP50 (from 32.6 to 31.7)?
2. There are relatively few comparative methods on the OV-LVIS benchmark.

---

> ### Author Response · Authors · 2024-11-22
> **Response to Reviewer s2q1 (part 1)**
>
> We extend our sincere gratitude for recognizing our method as reasonable and effective, addressing OVD with comprehensive experiments. Here's our response with some necessary explanations. If you have any further questions, please feel free to let us know.
>
> **Q1:** Performance drop of regional contrastive distillation loss.
>
> **Reply:** From line 2 to line 3, we can observe that adding regional contrastive loss alone decreases the effect of semantic guiding, but from line 2 to line 4 we can see that regional contrastive loss together with similarity classification improves the performance from 32.6 to 33.7, adding semantic guiding to line 4 further boosts to 38.0.
>
> |  #   | Def-DETR | Semantic Guiding | Regional-Contrastive Loss | Similarity Classification | $\text{AP}_{50}^{N}$ (%) |
> |------|----------|------------------|---------------------------|---------------------------|--------------------------|
> | 1    |✓         | ✗                | ✗                         | ✗                         | 25.4                     |
> | 2    |✓         | ✓                | ✗                         | ✗                         | 32.6                     |
> | 3    |✓         | ✓                | ✓                         | ✗    		             | 31.7                     |
> | 4    |✓         | ✗                | ✓                         | ✓    		             | 33.7                     |
> | 5    |✓         | ✓                | ✓                         | ✓    		             | 38.0                     |
>
> Line 3 and line 5 are models with identical weights (the model trained with distillation loss), the only difference lies in how they give classification predictions in inference, line 5 is a similarity-score classification while line 3 derives the prediction that incorporates logits score from classification branch.
>
> Since the same model presents disparate results with different post-process, we reckon the reason behind is: after training, the embedding space of CCKT-Det is pulled close to the hidden space of the teacher. Spcifically, added with textual features, the language queries are refined by decoder, then guided to be aligned with visual features encoded by the teacher. The tight integration of textual and visual features within the detection loop sets the optimization direction towards mimicking teacher on region level.
>
> Using vanilla post-process to give classification score fails to fully unleash the full potential of the model. Instead the classfication prediction should be follow the classification style of the teacher--CLIP performs zero-shot class predictions by measuring cosine similarities between images and texts. The similarity classfication postprocess completes the loop of our cyclic knowledge transfer, i.e., without introducing a gap between how student and teacher perform classification.
>
> **Q2:** Suggestions on supplementing introduction to the task setup, workflow details.
>
> **Reply:** Thanks for your suggestions, we add more descriptions mentioned above and refine them in the paper.
>
> **Q3:** Comparison or analysis of time complexity.
>
> **Reply:** Our CCKT-Det follows Deformable-DETR [R8], here we report training and inference time complexity of original Deformable-DETR, OV-DETR [R5] (another comparable Deformable-DETR style OVD detector) and our CCKT-Det.
>
> |  Model   | Epoches | Training GPU Hours | FLOPS (G) | Inference Time |
> |----------|----------|-------------------|-----------|----------------|
> | Deformable-DETR |  50  |   325 (NVIDIA Tesla V100)   |  173   |   0.31 (NVIDIA Tesla V100)  |
> | OV-DETR | 50  |   700 (NVIDIA RTX 3090)   |   --  |   0.66 (NVIDIA RTX 3090), 0.63 (NVIDIA Tesla V100)  |
> | CCKT-Det   | 36  |   324 (NVIDIA RTX 3090)  |  178   |    0.24 (NVIDIA RTX 3090)    |
>
> For training epoches and total GPU hours, our knowledge distillation based method requires less computation and training resources to converge.
>
> For computational complexity, we use the FLOPs and fvcore [R6] to get the statistics. OV-DETR [R5] suffers from repetitive per-class inference: both computation and memory consumption scales linearly with the vocabulary size [R7]. Since there is $C_{all} \times N_{queries}$ instead of only $N_{queries}$ needs to be fed into decoder each forward pass, we fail to get the FLOPs for OV-DETR with limited GPU memory at hand.
>
> For inference time, we follow OV-DETR to report the second per iteration. Our model is trained and tested on NVIDIA RTX 3090, all statistics related with NVIDIA Tesla V100 comes from [R5, R8].

---

> ### Author Response · Authors · 2024-11-22
> **Response to Reviewer s2q1(part2)**
>
> **Q4:** Report performance on base categories on COCO and LVIS.
>
> **Reply:** Thanks for your suggestions, we report them and provide relevant analysis.
>
> Results on the COCO dataset:
>
> | Model      |   $\text{AP}_{50}^{N}$ (%) | $\text{AP}_{50}^{B}$ (%) | $\text{AP}_{50}$ (%) | $\triangle \text{AP}_{50}$ (%) $\downarrow$|
> |------------|----------------------------|--------------------------|----------------------|-------------------------------|
> |ViLD-img    |    24.1    		          | 34.2                     | 31.6                 |10.1                           |
> |F-VLM       |    28.0    		          | -                        | 39.6                 |-                              |
> |OV-DETR     |    29.4    		          | 61.0                     | 52.7                 |31.6                           |
> |CORA        |    35.1    		          | 35.5                     | 35.4                 |0.4                            |
> |CoDet       |    30.6    		          | 52.3                     | 46.6                 |21.7                           |
> |CCKT-Det    |    38.0    		          | 34.1                     | 35.0                 |3.9                            |
> |CCKT-Det (SwinB)| 41.9    		          | 40.6                     | 40.9                 |1.3                            |
> |CCKT-Det++ (SwinB)    |   46.0           | 46.3                     | 46.2                 |0.3                            |
>
> We note that on base classes, our method is not as good as others. However, previous approaches might suffer from issue of overfitting on base classes. Here we define a new metric termed $\triangle \text{AP}_ {50}=|\text{AP}_ {50}^{B}-\text{AP}_ {50}^{N}|$ to quantitatively measure this phenomenon. The $\triangle \text{AP}_ {50}$ reflects the gap between performances on base and novel classes, open-vocabulary detection should achieve both high $\text{AP}_ {50}^{B}$ and $\text{AP}_ {50}^{N}$, but more importantly, a low $\triangle \text{AP}_ {50}$ to avoid skewed performance. As this new metric shows in the above table, the overfitting phenomenon is rampant especially on methods relying on extra image-caption pairs (CoDet) and self-training techniques (OV-DETR) since the generated pseudo boxes of base classes are more accurate than pseudo boxes of novel classes. Though our method is relatively low on base classes, we maintain a good balance between base and novel classes, achieving the lowest 0.3 $\triangle \text{AP}_ {50}$.

---

> ### Author Response · Authors · 2024-11-22
> **Response to Reviewer s2q1 (part3)**
>
> We also conduct a new experiment that trains CCKT-Det using data provided by OV-DETR (it includes generated pseudo boxes) to support our guess: we can observe performance of base objects surpassing novel objects by integrating more generated pseudo boxes as supervisions
>
> | Model       |$AP_{50}^{N}$ (%) | $AP_{50}^{B}$ (%) | $AP_{50}$ (%)     |$\triangle \text{AP}_{50}$ (%) $\downarrow$|
> |-------------|------------------|-------------------|-------------------|-------------------------------|
> |OV-DETR      |    29.4    		 | 61.0              | 52.7              |31.6                           |
> |CCKT-Det     |    38.0    		 | 34.1              | 35.0              |3.9                            |
> |CCKT-Det*    |   47.9 		     |  51.3             |    48.6           |3.4                            |
>
> If an open-vocabulary detector can only achieve superior performance on base categories but poor performance on novel classes, its effectiveness will be diminished, given that the evaluation of OVD essentially reflects not only AP but also the model's generalization ability to novel categories.
>
> Since the hidden space of VLMs exhibits minimal bias regardless of the division of base and novel classes on COCO dataset under OVD setting [R4], there is no distinct 'base' or 'novel' for VLMs, this balanced structural knowledge is effectively distilled to our detector. A similar trend can also be observed in other knowledge transfer or knowledge distillation-based methods, such as ViLD-img and CORA, achiving a relatively low $\triangle \text{AP}_ {50}$ compared to methods that utilize additional training data.
>
>
> In other words, the student performance is affected by teacher. If teacher has limitations, they could propagate to student. We can further observe that when we employ a more robust MLLM (GVT) to guide our model, the performance on both base and novel classes is boosted (c.f. to CCKT-Det and CCKT-Det++ in Tab. 1).
>
> To summarize, there is an inherent trade-off between base and novel categories, knowledge distillation based method could inherit both balanced semantics and limitations from the teacher. With the major evaluation metric for open-vocabulary detection being the novel AP, our method achieves superior performance while striking a good balance between base and novel classes.
>
> Results on LVIS dataset with $\text{AP}_ {c}$ and $\text{AP}_ {f}$:
>
> | Model      |   $AP_{r}$ (%) | $AP_{c}$ (%) | $AP_{f}$ (%) | $AP$ (%)       |
> |------------|----------------|--------------|--------------|----------------|
> |ViLD        |$16.3^{bbox}$   |$21.2^{bbox}$ |$31.6^{bbox}$ |$24.4^{bbox}$   |
> |OV-DETR     |$17.4^{mask}$   |$25.0^{mask}$ |$32.5^{mask}$ |$26.6^{mask}$   |
> |F-VLM       |$18.6^{mask}$   |-             |-             |$24.2$          |
> |CORA+       |$28.1^{bbox}$   |-             |-             |-               |
> |CoDet       |$29.4^{mask}$   |$39.5^{mask}$ | $43.0^{mask}$|$39.2^{mask}$   |
> |CoDet (EVA02-L-304M)  |$37.0^{mask}$ |$46.3^{mask}$ |$46.3^{mask}$ |$44.7^{mask}$ |
> |CCKT-Det++  |$32.8^{bbox}$   |$45.3^{bbox}$ |$45.5^{bbox}$ |$44.3^{bbox}$   |
>
> References:
>
> [R4] Open-Vocabulary Object Detection Using Captions. In CVPR 2021.
>
> [R5] Open-Vocabulary DETR with Conditional Matching. In ECCV 2022.
>
> [R6] fvcore https://github.com/facebookresearch/fvcore
>
> [R7] CORA: Adapting CLIP for Open-Vocabulary Detection With Region Prompting and Anchor Pre-Matching. In CVPR 2023.
>
> [R8] Deformable DETR: Deformable Transformers for End-to-End Object Detection. In ICLR 2021.

---

> > ### Comment · Reviewer_s2q1 · 2024-11-28
> >
> > I thank the authors for their response by conducting further experiments and analyses. The answers addressed most of my concerns, and I'd like to maintain my rating.

---

> ### Comment · Area_Chair_3RPh · 2024-11-27
>
> Dear reviewer,
>
> Today is the last day for reviewers to ask questions to authors. Did the authors' rebuttal address your concern? Do you have any additional questions?

---

### Official Review · Reviewer_B8e4 · 2024-11-03

**Soundness:** 2
**Presentation:** 3
**Contribution:** 2
**Rating:** 5
**Confidence:** 4

**Summary:**

In order to solve the problem of relying on a large amount of additional data for open vocabulary object detection, the authors propose CCKT-Det method. This method guides the learning of queries by pre-filtering and injecting semantic priors, and introduces regional contrast loss to improve the perception of queries to novel objects.

**Strengths:**

1. The frameworks are explicit and effectively illustrates the idea of the method.
2. This method has an improvement in the detection performance of novel classes.

**Weaknesses:**

1. The problem of the authors to solve and the motivation are both ambiguous. The authors point out that some methods rely on a large amount of extra caption data, but many methods can achieve high performance without it, such as the quoted OV-DQUO.
2. The authors introduce semantic prior, but do not clearly define the concept.
3. Figure 3 shows that the semantic prior needs to go through the VLM's text encoder, what is the difference between the semantic prior and the caption? And what is the difference between semantic priors and the prompts that currently popular? Neither of these issues is addressed by the authors.
4. The authors do not present the detection performance of the base class in the experiment.

**Questions:**

1. Clearly state the motivation for this paper.
2. It is necessary to give a clear definition of semantic prior and explain its difference from caption and prompts.
3. Please explain why the performance of the base class is not shown in the experiment.

---

> ### Author Response · Authors · 2024-11-22
> **Response to Reviewer B8e4 (part 1)**
>
> Thank you for acknowledging our framework as explicit and effective. Here's our response with some necessary explanations.
>
> **Q1:** The motivation of this paper.
>
> **Reply:** Thanks for your feedback. We clearly stated the motivation of our work as bellowing. Open Vocabulary Object Detection (OVD) aims to identify objects from novel categories that were not encountered during the training process. According to the training data, OVD methods can be mainly grouped into [R1]: 1) methods that utilize additional image-caption pairs beyond the detection training data; 2) methods that employ self-training strategy and generate additional pseudo labels for both base and novel classes on detection training data; 3) methods that rely on knoledge distillation from teacher VLMs. Our approach falls into the third group.
>
> While methods that leverage 1) additional image-caption pairs or 2) pseudo labels can enhance the performance of open-vocabulary detectors, they often exhibit low data efficiency and incur high computational costs. Our proposed cyclic contrastive knowledge transfer seeks to optimize the knowledge transfer efficacy of the third knowledge distillation group without the need for supplementary data. Existing knowledge distillation approaches primarily focus on learning from the visual knowledge embedded in the teacher model. In contrast, we advocate for a cyclic learning process that incorporates both text and visual knowledge from the teacher by enveloping the detector in this cycle, while excluding the object queries, as they serve as inputs to the detector. This methodology significantly enhances cross-modal learning, as evidenced by our experimental results.
>
> It is also worth noting that OV-DQUO adopts an iterative training strategy that gradually generates and adds unknown object proposals into the training set, hence it aligns more closely with pseudo-labeling group.
>
> References:
>
> [R1] A Survey on Open-Vocabulary Detection and Segmentation: Past, Present, and Future. TPAMI 2024.
>
> **Q2:** The definition of semantic priors and its difference between captions and prompts.
>
> **Reply:** Sorry for the confusion. The semantic priors denotes the presence of a specific concept in current image. In existing OVD, the input texts to text encoder are typically category names from datasets, and they are fixed for each image. In contrast, we filter out non-existent concepts based on either VLMs, i.e., CLIP, or MLLMs, i.e., GVT. Hence the input texts are dynamic for each image. The similarity logits w.r.t. categories that are below threshold *ρ* are filtered out to ensure that only highly confident classes are identified as present ones. This avoids novel classes being overwhelmed by many non-existent base classes as our ablation in Sec. 4.1 confirms.
>
> The semantic prior embeddings are text features of those remaining classes (above the threshold *ρ*) encoded by CLIP text encoder via filling their category names into template prompts. These embeddings are added to the learnable object queries to form the language queries and forwarded to decoder for box regression and classification.
>
> **Difference between semantic priors and captions or prompts:** Prompts can be categorized into template prompts (e.g., "a photo of [CLASS].") and soft continuous prompts by prepending several learnable vectors before the [CLASS] token. We use semantic priors to resolve the concept existence problem stated in Sec. 3.2 and apply the template prompts to filtered classes.

---

> ### Author Response · Authors · 2024-11-22
> **Response to Reviewer B8e4 (Part 2)**
>
> **Q3:** The performance on base classes
>
> **Reply:** Current setting of open-vocabulary detection is to train the detector on base categories $C_B$ and test on both base and novel categories $C_N$, where $C^B \bigcap C^N = \emptyset$. OVD mainly uses the AP for novel classes as the evaluatation metric, and we follow standard reporting practices [R2-R4] in literature that includes AP on novel and all categories.
>
> Here we include the AP for both base and novel classes for a complete reference, and provide relevant analysis:
>
> Results on COCO dataset:
>
> | Model      |   $\text{AP}_{50}^{N}$ (%) | $\text{AP}_{50}^{B}$ (%) | $\text{AP}_{50}$ (%) | $\triangle \text{AP}_{50}$ (%) $\downarrow$|
> |------------|----------------------------|--------------------------|----------------------|-------------------------------|
> |ViLD-img    |    24.1    		          | 34.2                     | 31.6                 |10.1                           |
> |F-VLM       |    28.0    		          | -                        | 39.6                 |-                              |
> |OV-DETR     |    29.4    		          | 61.0                     | 52.7                 |31.6                           |
> |CORA        |    35.1    		          | 35.5                     | 35.4                 |0.4                            |
> |CoDet       |    30.6    		          | 52.3                     | 46.6                 |21.7                           |
> |CCKT-Det    |    38.0    		          | 34.1                     | 35.0                 |3.9                            |
> |CCKT-Det (SwinB)| 41.9    		          | 40.6                     | 40.9                 |1.3                            |
> |CCKT-Det++ (SwinB)    |   46.0           | 46.3                     | 46.2                 |0.3                            |
>
> We note that on base classes, our method is not as good as others. However, previous approaches might suffer from issue of overfitting on base classes. Here we define a new metric termed $\triangle\text{AP}_ {50}=|\text{AP}_ {50}^{B}-\text{AP}_ {50}^{N}|$ to quantitatively measure this phenomenon. The $\triangle \text{AP}_ {50}$ reflects the gap between performances on base and novel classes, open-vocabulary detection should achieve both high $\text{AP}_ {50}^{B}$ and $\text{AP}_ {50}^{N}$, but more importantly, a low $\triangle \text{AP}_ {50}$ to avoid skewed performance. As this new metric shows in the above table, the overfitting phenomenon is rampant especially on methods relying on extra image-caption pairs (CoDet) and self-training techniques (OV-DETR) since the generated pseudo boxes of base classes are more accurate than pseudo boxes of novel classes. Though our method is relatively low on base classes, we maintain a good balance between base and novel classes, achieving the lowest 0.3 $\triangle \text{AP}_ {50}$.

---

> ### Author Response · Authors · 2024-11-22
> **Response to Reviewer B8e4 (part 3)**
>
> We also conduct a new experiment that trains CCKT-Det using data provided by OV-DETR (it includes generated pseudo boxes) to support our guess: we can observe performance of base objects surpassing novel objects by integrating more generated pseudo boxes as supervisions
>
> | Model       |$AP_{50}^{N}$ (%) | $AP_{50}^{B}$ (%) | $AP_{50}$ (%)     |$\triangle \text{AP}_{50}$ (%) $\downarrow$|
> |-------------|------------------|-------------------|-------------------|-------------------------------|
> |OV-DETR      |    29.4    		 | 61.0              | 52.7              |31.6                           |
> |CCKT-Det     |    38.0    		 | 34.1              | 35.0              |3.9                            |
> |CCKT-Det*    |   47.9 		     |  51.3             |    48.6           |3.4                            |
>
> If an open-vocabulary detector can only achieve superior performance on base categories but poor performance on novel classes, its effectiveness will be diminished, given that the evaluation of OVD essentially reflects not only AP but also the model's generalization ability to novel categories.
>
> Since the hidden space of VLMs exhibits minimal bias regardless of the division of base and novel classes on COCO dataset under OVD setting [R4], there is no distinct 'base' or 'novel' for VLMs, this balanced structural knowledge is effectively distilled to our detector. A similar trend can also be observed in other knowledge transfer or knowledge distillation-based methods, such as ViLD-img and CORA, achiving a relatively low $\triangle \text{AP}_{50}$ compared to methods that utilize additional training data.
>
>
> In other words, the student performance is affected by teacher. If teacher has limitations, they could propagate to student. We can further observe that when we employ a more robust MLLM (GVT) to guide our model, the performance on both base and novel classes is boosted (c.f. to CCKT-Det and CCKT-Det++ in Tab. 1).
>
> To summarize, there is an inherent trade-off between base and novel categories, knowledge distillation based method could inherit both balanced semantics and limitations from the teacher. With the major evaluation metric for open-vocabulary detection being the novel AP, our method achieves superior performance while striking a good balance between base and novel classes.
>
> Results on LVIS dataset with $\text{AP}_ {c}$ and $\text{AP}_ {f}$:
>
> | Model      |   $AP_{r}$ (%) | $AP_{c}$ (%) | $AP_{f}$ (%) | $AP$ (%)       |
> |------------|----------------|--------------|--------------|----------------|
> |ViLD        |$16.3^{bbox}$   |$21.2^{bbox}$ |$31.6^{bbox}$ |$24.4^{bbox}$   |
> |OV-DETR     |$17.4^{mask}$   |$25.0^{mask}$ |$32.5^{mask}$ |$26.6^{mask}$   |
> |F-VLM       |$18.6^{mask}$   |-             |-             |$24.2$          |
> |CORA+       |$28.1^{bbox}$   |-             |-             |-               |
> |CoDet       |$29.4^{mask}$   |$39.5^{mask}$ | $43.0^{mask}$|$39.2^{mask}$   |
> |CoDet (EVA02-L-304M)  |$37.0^{mask}$ |$46.3^{mask}$ |$46.3^{mask}$ |$44.7^{mask}$ |
> |CCKT-Det++  |$32.8^{bbox}$   |$45.3^{bbox}$ |$45.5^{bbox}$ |$44.3^{bbox}$   |
>
> References:
>
> [R2] F-VLM: Open-Vocabulary Object Detection upon Frozen Vision and Language Models. In ICLR 2023.
>
> [R3] Region-centric Image-Language Pretraining for Open-Vocabulary Detection. In ECCV 2024.
>
> [R4] Open-Vocabulary Object Detection Using Captions. In CVPR 2021.

---

> ### Comment · Area_Chair_3RPh · 2024-11-27
>
> Dear reviewer,
>
> Today is the last day for reviewers to ask questions to authors. Did the authors' rebuttal address your concern? Do you have any additional questions?

---

> > ### Comment · Reviewer_B8e4 · 2024-11-27
> >
> > Thanks for the feedback. My concerns have been addressed. I will maintain my current rating.

---

> > > ### Author Response · Authors · 2024-12-02
> > > **Response to Reviewer B8e4 (part 5)**
> > >
> > > Thanks for your attention and feedback on our work. We are pleased to hear that your concerns have been addressed. We understand that the scoring process is rigorous and fair, but we hope you might consider whether our responses and clarifications could positively influence your overall evaluation of the paper.
> > >
> > > If you have any further questions or uncertainties, please feel free to reach out. We would be more than happy to provide additional clarifications. Thank you once again for your time and suggestions.

---

> ### Author Response · Authors · 2024-11-27
> **Response to Reviewer B8e4 (part 4)**
>
> Thanks for your response and for taking the time to consider our rebuttal. We appreciate your acknowledgment that your concerns have been addressed. However, we would like to understand your reasoning behind maintaining a negative rating. If there are specific aspects of our work that you believe still require improvement or clarification, we would be grateful for your feedback. Our goal is to enhance the quality of our paper, and your insights would be invaluable in this process. Thank you once again for your time and consideration.

---

### Official Review · Reviewer_6x5S · 2024-11-05

**Soundness:** 3
**Presentation:** 3
**Contribution:** 3
**Rating:** 8
**Confidence:** 4

**Summary:**

This paper introduces CCKT-Det++, an open-vocabulary object detection framework designed to detect novel object categories without relying on extensive additional supervision. The approach leverages semantic priors as guidance and a regional contrastive knowledge distillation loss to improve the model's detection capability for novel classes. CCKT-Det++ utilizes visual-semantic embeddings from both VLMs and MLLMs to dynamically transfer knowledge from both image and text encoders, aligning the detector’s latent space with meaningful, structured knowledge.

**Strengths:**

This work is easy to follow. Most of the used techniques are correct.

**Weaknesses:**

1. While CCKT-Det++ performs well by leveraging stronger teacher models, it is heavily reliant on the quality and alignment of these teacher models. If the teacher model has limitations or biases, these could propagate to CCKT-Det++, affecting performance and potentially introducing unintended biases.

2. Although the student model maintains moderate parameters, using stronger teacher models during training and inference can introduce significant computational costs. Scaling the teacher models effectively requires access to substantial computational resources, which could limit practical applications for researchers without high-performance infrastructure.

3. The approach relies on pseudo annotations, which are less accurate than human annotations. If the pseudo labels are of low quality, they could introduce noise into the training process, possibly harming model performance on more challenging or nuanced object classes.

4. While semantic priors improve performance on novel objects, there is a risk that the model might over-rely on these priors, leading to reduced flexibility in identifying objects that fall outside its learned semantic scope. This dependency may affect its generalization to truly unseen classes in new environments.

5. The method heavily relies on CLIP for both semantic priors and feature extraction, making it vulnerable to limitations inherent in the CLIP model. For instance, CLIP's biases in language and visual associations could impact detection accuracy and lead to incorrect classifications in culturally or contextually sensitive settings.

**Questions:**

Most concerns are listed on above boxes. Hungarian matching based on CLIP embeddings is central to your contrastive knowledge transfer scheme, aligning regional embeddings with the teacher’s visual-semantic space. However, given that CLIP embeddings may not fully capture object-specific regional nuances, why would this Hungarian matching approach be expected to significantly improve performance?

---

> ### Author Response · Authors · 2024-11-22
> **Response to Reviewer 6x5S**
>
> Thanks for your valuable comments and constructive suggestions. The following is our response with some necessary explanations. If you have any further questions, please feel free to let us know.
>
> **Q1:** How would hungarian matching based on CLIP embeddings expected to significantly improve performance.
>
> **Reply:**
> In our proposed contrastive knowledge transfer cycle, query embeddings are designed to align with region-level visual features provided by the teacher model. The contrastive loss establishes the optimization target to improve this alignment. To provide a more precise response, let’s consider this from the perspective of positive pairs in the contrastive loss framework.
>
> **Teacher Embeddings:** In various region-level contrastive learning frameworks for object detection tasks, positive instances are generated in a self-supervised manner. These instances are derived from augmented views of the same object, characterized by variations in scale and location. In our method, positive instances are sampled from a teacher model using an object-specific feature cache. We utilize a visual encoder to extract features from cropped images, which are obtained from the bounding boxes of base categories. These region-level features are subsequently stored in a cache and sampled during each training iteration. For further details, please refer to Figure 3 in our main paper.
>
> **Student Embeddings:** We enhance query embeddings through the integration of semantic priors. In contrast to the original queries in standard DETR, which are initialized to zero embeddings with learnable weights, this approach enables the queries to demonstrate increased differentiation during the Hungarian matching phase. Consequently, queries that are more likely to correspond to valid objects are selected by the Hungarian matcher and regarded as positive instances from the student model. In essence, the incorporation of semantic guidance enriches the semantic priors for the queries during the subsequent Hungarian matching, thereby facilitating the optimization process.
>
> **Q2:** Limitations or biases in the teacher model may propagate to CCKT-Det.
>
> **Reply:** Yes, we agree with your opinion. We reckon that CCKT-Det excels by leveraging advanced teacher models for semantic priors and feature extraction, it depends on their quality. Consequently, any limitation or biase inherent in these sources can affect our accuracy and introduce biases. However most open-vocabulary detection methods rely on teacher VLMs to achieve a good visual-semantic alignment, hence we leave this for future work.
>
> **Q3:** Computational costs for researchers lacking high-performance infrastructure.
>
> **Reply:** Leveraging VLMs/MLLMs indeed requires high-performance infrastructure. Scaling the teacher models provides an alternative to directly scaling the backbone of the student model for improved performance. The latter approach incurs increasing FLOPs with each forward pass, however, for researchers without high-performance devices, features from the teacher can be extracted once, saved as a .pkl file, and then loaded as needed.
>
> **Q4:** Pseudo labels incur noise.
>
> **Reply:** CCKT-Det utilize pseudo labels to determine whether a concept exist within one image. To mitigate noise, ground-truth annotations of base classes are employed during the training process. Pseudo labels are only utilized during the inference phase.
>
> **Q5:** Semantic priors not generalizable to real-world applications.
>
> **Reply:** Semantic priors enhance the model to detect novel objects within the learned semantic scope of teacher. Current evaluation settings and performance already demonstrate the generalization ability of CCKT-Det to novel concepts. While promising, the open-vocabulary detection community faces challenges in more fine-grained and unexpected scenarios, as the pretraining corpus of teacher VLMs may lack some "out-of-distribution" domain-specific data.

---

> ### Comment · Area_Chair_3RPh · 2024-11-27
>
> Dear reviewer,
>
> Today is the last day for reviewers to ask questions to authors. Did the authors' rebuttal address your concern? Do you have any additional questions?

---

> > ### Comment · Reviewer_6x5S · 2024-11-27
> > **Post-Rebuttal**
> >
> > Thanks for the feedback. My concerns have been addressed. I will maintain my current rating.

---

> > > ### Author Response · Authors · 2024-11-27
> > > **Thanks for your reply**
> > >
> > > Thank you for your acknowledgment of our work and for the efforts you put into this review process.

---

### Author Response · Authors · 2024-11-22
**General response**

We thank all reviewers for their valuable comments and constructive suggestions. We have made the revisions to the main paper accordingly, and the revised part is marked by blue. The main revisions are summarized as follows:

1. Added more explanations about ablation experiments; added more clarifications and descriptions when necessary.
2. Some minor modifications over clerical error.

The details of the revisions and other supplemented experiment results are referred to the following official comments.

---

### Author Response · Authors · 2024-11-25
**General response 2**

All authors would like to thank the reviewers again for their valuable efforts. As the discussion period is coming to an end, if you have any further comments or suggestions, please feel free to let us know. Thank you.

---

### Meta-Review · Area_Chair_3RPh · 2024-12-25

**Metareview:**

This paper was reviewed by five experts in the field. The paper received mixed review ratings of 8, 5, 6, 5, and 6. The AC read the paper, reviews, and rebuttal carefully.

Reviewer B8e4 and kmme are negative about the paper. Both reviewers asked for results on base classes. Reviewer B8e4 also had some clarification questions. During the rebuttal, reviewer B8e4 said their concerns had been addressed, but they still kept the original rating without further explanation. In this case, the AC considered the clarification questions raised by reviewer B8e4 were addressed. The authors' response included detailed results on base classes, and the AC believes they sufficiently addressed the concerns. These results should be incorporated into the final paper.

Therefore, the decision is to recommend the paper for acceptance. The reviewers did raise some valuable suggestions in the discussion that should be incorporated in the final camera-ready version of the paper. The authors are encouraged to make the necessary changes to the best of their ability.

**Additional Comments On Reviewer Discussion:**

During the rebuttal, three reviewers were positive about the paper and the other two were negative (reviewer B8e4 and kmme). Both reviewer B8e4 and kmme asked for results on base classes. Reviewer B8e4 also had some clarification questions. During the rebuttal, reviewer B8e4 said their concerns had been addressed, but they still kept the original rating without further explanation. In this case, the AC considered the clarification questions raised by reviewer B8e4 were addressed. The authors' response included detailed results on base classes, and the AC believes they sufficiently addressed the concerns.

---

### Decision · Program_Chairs · 2025-01-22

Accept (Poster)